# New Biomarkers and Treatment Advances in Triple-Negative Breast Cancer

**DOI:** 10.3390/diagnostics13111949

**Published:** 2023-06-02

**Authors:** Brahim El Hejjioui, Salma Lamrabet, Sarah Amrani Joutei, Nadia Senhaji, Touria Bouhafa, Moulay Abdelilah Malhouf, Sanae Bennis, Laila Bouguenouch

**Affiliations:** 1Biomedical and Translational Research Laboratory, Faculty of Medicine and Pharmacy, Sidi Mohamed Ben Abdellah University, Fez 30050, Morocco; 2Department of Medical Genetics and Oncogenetics, HASSAN II University Hospital, Fez 30050, Morocco; 3Department of Radiotherapy, HASSAN II University Hospital, Fez 30050, Morocco; 4Faculty of Sciences, Moulay Ismail University, Meknès 50000, Morocco; 5Obstetrics Gynecology Department, HASAN II University Hospital, Fez 30050, Morocco

**Keywords:** triple-negative breast cancer (TNBC), molecular subtype, biomarkers, omics, neoadjuvant treatments

## Abstract

Triple-negative breast cancer (TNBC) is a specific subtype of breast cancer lacking hormone receptor expression and HER2 gene amplification. TNBC represents a heterogeneous subtype of breast cancer, characterized by poor prognosis, high invasiveness, high metastatic potential, and a tendency to relapse. In this review, the specific molecular subtypes and pathological aspects of triple-negative breast cancer are illustrated, with particular attention to the biomarker characteristics of TNBC, namely: regulators of cell proliferation and migration and angiogenesis, apoptosis-regulating proteins, regulators of DNA damage response, immune checkpoints, and epigenetic modifications. This paper also focuses on omics approaches to exploring TNBC, such as genomics to identify cancer-specific mutations, epigenomics to identify altered epigenetic landscapes in cancer cells, and transcriptomics to explore differential mRNA and protein expression. Moreover, updated neoadjuvant treatments for TNBC are also mentioned, underlining the role of immunotherapy and novel and targeted agents in the treatment of TNBC.

## 1. Introduction

In the physiological and normal state of the breast, the mammary gland is a branched tubular gland constituted by galactophorous ducts. These latter progressively branch into the surrounding fibro-adipose tissue to terminate in lobules. The ductal or lobular nature is limited by a basal membrane, ensuring an interface with the connective tissue as well as the endothelial and contractile cells. The plasticity of this tubular mammary epithelial gland implies the succession of apoptosis, proliferation, differentiation, and maturation phenomena. It allows renewal of the different types of mammary cells, but subjects them to possible accidents during cell division [1].

Accumulation of genomic and epigenomic aberrations enables the development of cancerous aspects, such as reduced apoptosis, uncontrolled proliferation, increased motility, and angiogenesis. Identification of the genes (oncogenes and tumor suppressor genes) is important to understand the mechanisms of cancer development and to guide future improvements in cancer diagnosis and treatment [2,3].

Mammography is the gold-standard screening tool, but its performance is poorer in younger women. As an alternative for this group, thermal imaging or thermography is safer, more effective, and non-invasive [4]. Subsequent to the mammogram, if an abnormality is detected, additional examinations are required to confirm the existence of the tumor, including ultrasound, magnetic resonance imaging (MRI), and biopsy [5].

Breast cancers can be classified into subgroups according to histological type and grade [6].

Histological type is based on a variety of criteria, such as tumor cell type, extracellular secretion, architectural features, and immunohistochemical profile. Twenty histological types of invasive breast cancers can be differentiated, including infiltrating duct carcinomas, no special type (IDC-NST), and invasive lobular carcinoma (ILC) [7].

On the other hand, the grade reflects the aggressiveness of the tumor [8]. The Scarff–Bloom–Richardson classification system (SBR) predicts three grades (I-III) based on the analysis of three parameters: tubule formation, nuclear pleomorphism, and mitotic count. Each feature is scored 1, 2 or 3 and the final SBR score is between 3 and 9 [9]. SBR classification is used to provide an accurate diagnosis to predict the behavior of the tumor [7,10].

The expressions of ER, PR, and HER2 have been identified as prognostic markers and important predictive factors for hormone and anti-HER2-targeted therapy. ER/PR-positive cancers are expressed in ∼75% of all breast cancers, which are generally low-grade and less aggressive. Overall, 15% of breast cancers overexpress HER2. HER2 overexpression is associated with an aggressive clinical outcome and poor prognosis, but it is also predictive of response to anti-HER2 targeted therapies. The remaining 10–15% are characterized by absence of expression of the three markers (ER-, PR-, HER2-). This is called triple-negative breast cancer (TNBC); this group is associated with a high grade and a poor prognosis [7,11,12].

**EMAN classification** has revealed further complexities with the emergence of six different intrinsic subtypes, namely luminal A, luminal B, HER2-enriched, normal-like, basal-like, and claudin-low. This classification is based on the presence or absence of the three primary markers ER, PR, and HER2; the basal marker (CK5/6, EGFR); and the proliferation index Ki-67 [13]. TNBC classification will guide clinicians to the right treatment, as each subtype responds differently to treatment. Several targeted therapies have been approved, including PARP inhibitors, olaparib (approved in 2018), and talazoparib (approved in 2019), for patients with *BRCA* mutations. Other therapies for TNBC have been approved by FDA, such as the programmed cell death ligand 1 (PD-L) inhibitor atezolizumab and the antibody drug conjugate (ADC) sacituzumab govitecan. In addition, promising targeted therapies are in Phase I/II trials, mainly in combination with standard of care and as single agents [14,15].

The luminal type corresponds to tumors with ER expression and is classified into **luminal A** and **luminal B** according to the presence of HER2 expression. Luminal A is HER2-negative, while luminal B is HER2-positive and associated with a worse prognosis.

**The HER2-enriched breast cancer subtype** is characterized by the absence of ER and PR expression but has positive HER2. However, this subtype is often successfully treated with *trastuzumab*, which improves the prognosis.

**Normal-type tumor cells** are characterized by the absence of HER2 and KI67 expression. This subtype has a low percentage of tumor cells and a lack of expression of proliferation genes.

The **basal-like subtype**, called triple-negative breast cancer (TNBC), is negative for the expression of ER, PR, and HER2 proliferation genes. The term “basal-like” is due to the similarity in the expression of the following basal markers: epidermal growth factor receptor (EGFR), CK5/6, CK14, and CK17. TNBC is more aggressive than the above-mentioned subtypes. The aggressivity of TNBC is explained by the lack of response to targeted or hormone therapy. Therefore, cytotoxic drugs are the only option for the treatment of TNBC. This subtype is common in women with a mutation in the breast cancer gene *BRCA*.

The **low-claudin breast cancer subtype** is also known as triple-negative breast cancer. As the name suggests, it is characterized by low expression of claudins (3, 4, 7), E-cadherin, mucin-1, and epithelial cell adhesion molecules (EpCAM) [13,16].

The advancement of translational informatics, genomics and proteomics makes it possible to decode breast cancer heterogeneity in a clinically applicable manner [17]. Therefore, precision medicine is being implemented through the development of targeted therapies to match TNBC subtypes [14].

Current molecular classification of breast cancer uses intrinsic PAM50 subtyping determined by the *Prosigna Breast Cancer Prognostic Gene Signature Assay* (formerly called the PAM50 test). Classification is based on the analysis of 50 genes in a surgically resected breast cancer sample. The PAM50 test classifies a tumor as one of four intrinsic subtypes (luminal A, luminal B, HER2-enriched, or basal-like). In addition to this classification, the test generates an individualized score estimating a patient’s probability of disease recurrence [18,19].

TNBC is very heterogeneous. Over the last 20 years, much effort has been devoted to a more exhaustive classification of TNBC. The aim is to stratify patients for better diagnosis and thus for appropriate management. Nowadays, efforts are focused largely on understanding triple-negative breast cancer (TNBC) biology, since it is the most clinically aggressive group as (i) there is no expression of three biomarkers (ER, PR and Her2), (ii) it affects the youngest women, and (iii) current treatment of TNBC relies largely on chemotherapy and radiotherapy [17,20].

The heterogeneity of TNBC was first studied by Lehmann et al. [21] Based on gene expression profiling of 2188 genes in 587 TNBC patients, they classified TNBC into six new groups, namely basal-like 1 (BL1), basal-like 2 (BL2), immunomodulatory (IM), luminal androgen receptor (LAR), mesenchymal stem-like (MSL), and mesenchymal (M). 

BL1 (22%) and BL2 (12%) are characterized by high levels of Ki67 and DNA damage response. The IM subtype (18%) showed characteristics of the basal type with activation of IFNα and IFNγ signaling. The IM subtype is also characterized by high gene expression of cytotoxic T-cell-associated protein 4. The subtypes M (21%) and MSL (10%) are associated with EGFR, calcium, MAPK, and PI3K signaling deregulation. In the LAR subtype (9%), an approximately 10-fold increase in androgen receptor (AR) expression was observed compared with the other subtypes. Activation of various pathways, such as steroid synthesis and FOXA1 and ERBB signaling, was also observed in this subtype [21].

Honma and colleagues have shown that TNBC can be classified into five categories. This subclassification can be very useful, since each subtype not only represents specific molecular features, but also suggests dedicated treatment options. These are luminal AR (LAR), immunomodulatory (IM), basal-like 1 (BL1), basal-like 2 (BL2), and mesenchymal/mesenchymal stem-like (M/MSL). (Table 1) [22].

TNBC is a collection of distinct diseases that share the absence of expression of ER and PR and no amplification of HER2 but are very different in terms of histological features, genomic characteristics, clinical behavior, and response to treatment. TNBC stratification into biological and/or clinical subgroups will be essential for the personalization of therapy [23].

## 2. TNBC Characteristic Biomarkers

### 2.1. Regulators of Cell Proliferation and Migration and Angiogenesis

#### 2.1.1. Epidermal Growth Factor Receptor (EGFR)

EGFR is a receptor belonging to the ErbB family. It is a glycoprotein, including an extracellular part for ligand binding (EGF), a transmembrane domain, and an intracellular part carrying tyrosine kinase activity. It is coded by the *EGFR* gene, located on the arm of chromosome 7 (7p11.2). After ligand activation, the inactive monomer receptor dimerizes, followed by autophosphorylation of the intracellular domain, leading to the activation of multiple signal transduction pathways, such as RAS-RAF-MEK-ERK, PI3K-AKT-mTOR, and SRC-STAT3. Hence, the EGFR receptor is associated with regulation of the cell cycle, proliferation, differentiation, and survival. EGFR is also known to translocate into the nucleus, where it is involved in transcriptional regulation, DNA replication, and DNA repair [23,24]. 

A number of studies have reported that EGFR is more frequently overexpressed in TNBC compared with other subtypes of breast cancer. Positive expression of EGFR has been identified as a poor prognostic factor for TNBC. This is mainly due to amplification or mutations of its gene. EGFR expression, gene amplification, and mutation status have been extensively studied in TNBC. In fact, expression has been reported in about 60% of TNBC, whereas EGFR amplification has been identified in only 2% to 24% of cases. Furthermore, a high copy number has been found in 8% to 27% of TNBC samples. However, EGFR mutations are a rare event, occurring in only about 11% of samples [24].

Several clinical trials have evaluated the efficacy of targeting EGFR. There are mainly two therapeutic approaches: tyrosine kinase inhibitors (EGFR-TKIs) and monoclonal antibodies (mAbs) [25]. EGFR-TKIs block the binding of adenosine 5′ triphosphate to the intracellular catalytic domain of EGFR, whereas anti-EGFR mAbs target the extracellular domain. In both scenarios, mAbs and EGFR-TKIs inhibit EGFR activation and, consequently, eliminate its downstream signal transduction. These therapeutic procedures can be carried out alone or in combination with chemotherapy. However, results so far have been disappointing. Nevertheless, a small group of TNBC patients respond positively to anti-EGFR therapy, hence the importance of identifying EGFR activation [26,27,28].

#### 2.1.2. Fibroblast Growth Factor Receptor (FGFR)

Fibroblast growth factor receptors are a family of four highly conserved transmembrane receptor tyrosine kinases, including FGFR1, FGFR2, FGFR3, and FGFR4. Membrane-bound FGFRs are activated by the binding of ligand FGFs to their extracellular domains. The FGF family includes secreted FGFs (FGF1, FGF4, FGF7, FGF8, FGF9, and the FGF15/19 subfamilies) and intracellular FGFs (the FGF11 subfamily). They are implicated in different physiologic functions, including embryonic development, regulation of cell proliferation and differentiation (MAPK, PI3K-AKT), angiogenesis (MAPK-kinase p38; PI3K-AKT-FOXO1, TSC2), apoptosis (MAPK- JNK; PI3K-AKT- FOXO1, TSC2), metabolism, and motility (PLCγ -IP3- DAG, PKC). Therefore, aberrant activation of the FGF/FGFR signaling pathway has been consistently associated with human cancers as well as developmental defects and genetic and metabolic disorders [24,29,30].

The *FGFR1* gene, located on chromosome 8p11–12, is mutated in around 15% of breast cancer patients and 7% of TNBC patients. This specific region is generally amplified, with the 11q12–14 region containing other oncogenes with a determinant role in breast cancer progression, such as *CCND1*, *FGF3*, *FGF4*, and *FGF19*. *FGFR1* amplification is an independent negative prognostic factor in TNBC correlated with low survival rates [31]. 

*FGFR2* amplification (10q26.13) is also a common *FGFR* aberration, occurring in 5–10% of breast cancers and 4% of TNBCs. In TNBC, *FGFR2* amplification has been correlated with robust activation of cellular transformation and resistance to FGFR inhibitors [32].

The *FGFR3* gene (4p16.3) is mutated in a very low percentage of breast cancer patients, particularly in the extracellular and the transmembrane protein domains, and rarely within the kinase domain. Gene fusions have also been reported. For instance, *FGFR3-TACC3* gene fusion was found to be highly expressed in TNBC cell lines, resulting in autophosphorylation of FGFR3. Moreover, amplification of the FGFR3 gene has been characterized in less than 1% of breast cancer patients [33].

Mutations in the *FGFR4* gene (5q35.2) lead to modifications at K535 and E550 in the kinase domain of the FGFR4 protein in breast cancer patients, causing protein autophosphorylation and activation. These specific mutations are mostly found in metastatic breast cancers, implying a positive correlation between FGFR4 signaling and metastatic breast cancer. On the other hand, FGFR4 gene amplification is detected in only 2.3% of all breast cancers. Additionally, the FGFR4-R388 allele is associated with tumorigenesis, cell motility, and immune evasion [34,35].

FGFR inhibitors are a novel class of drugs in breast cancer. Indeed, FGFR inhibitors can be classified into three distinct groups: the first type, preventive therapy, is given before a breast cancer diagnosis in high-risk patients; the second is adjuvant therapy, having as its main function the reduction of the risk of metastases after surgery, especially in early breast cancer patients; the third group are administrated in the context of established metastatic breast cancer in order to slow the progression of the disease and consequently improve survival [29,36,37].

#### 2.1.3. Vascular Endothelial Growth Factor (VEGF)

Vascular endothelial growth factor (VEGF), responsible for mediating angiogenesis signaling, is essential in the process of tumor growth, and spreads particularly by transporting nutrients and oxygen. The production of VEGF by tumor cells is stimulated by various signals, such as inflammation and hypoxia [24].

The *VEGF* gene is located on the 6p21.3 chromosome. It is part of the *VEGF/PDGF* gene family. In humans, the VEGF family includes six vascular endothelial growth factors, VEGFA to VEGFE, to which must be added placental growth factor (PGF). There are three corresponding tyrosine kinase receptors: VEGFR1, VEGFR2, and VEGFR3. VEGF-A is the most important stimulator of angiogenesis. It comprises eight exons separated by seven introns. Alternative mRNA splicing leads to the production of several isoforms with different biological properties. The most common form, VEGFA 165, lacks the exon 6 product [38,39].

Many researchers have identified strong VEGF expression in approximately 30–60% of patients with TNBC compared with non-TNBC patients. Generally, this high expression is associated with poor prognosis and shorter overall survival. 

The purpose of anti-VEGF therapies is to inhibit angiogenesis by blocking VEGF or its receptors (VEGFRs). Bevacizumab, an anti-VEGF antibody, is the only antiangiogenic medicine currently approved by the NCCN Clinical Practice Guidelines in Oncology for treatment of TNBC patients. In fact, several clinical studies with bevacizumab have demonstrated a significant improvement in overall progression-free survival. However, no significant improvement was reported in overall survival. Moreover, numerous side effects of these antiangiogenic drugs, such as malignant hypertension and cutaneous, renal, hepatic, and hematological toxicities, usually mask the possible positive effects of this therapy [40,41]. Consequently, due to its limited effect on overall patient survival, bevacizumab is not currently recommended in the first-line setting to treat metastatic TNBC [42]. Therefore, the development of new antiangiogenic drugs, especially small molecules with low toxicity, is essential for treatment of TNBC patients.

Currently, there is an ongoing, prospective, randomized clinical trial including 128 TNBC patients [43]. This research is evaluating the efficacy and the safety of albumin-bound paclitaxel combined with antiangiogenic agents in first-line treatment of relapsed or metastatic TNBC. The study started in January 2022 for a period of 2 years. 

### 2.2. Proteins Regulating Apoptosis

#### 2.2.1. Protein p53

Tumor suppressor p53 is encoded by the gene *TP53*, located on chromosome 17, and has 13 exons. This protein is highly involved in sustaining homeostasis and cell integrity by acting as “the genome guardian”. The transactivation of two wild-type p53 domains (TAD1 and TAD2) implicates cell arrest and DNA repair by generating local senescence and inhibiting VEGF-associated angiogenesis, in addition to preventing metastasis and tumor growth [44,45]. The transcription of pro-apoptotic proteins is linked to the relocation of wtp53 to the mitochondria, which causes membrane permeability to be damaged [46]. Alterations in p53 functions are generally associated with mutations that cause a change in some amino acids, causing p53/DNA association to be altered. Nevertheless, the following post-transduction mechanisms could be the cause of the misleading reconnaissance between the mature p53 form and the promotor binding regions (i.e., promoter selection) [44,47]. Practically, 65% to 80% of TNBC cases present *TP53* mutations. These are frequent in the p53/DNA binding region, leading to the worst prognosis [48,49,50]. This latter is also noted in a specific polymorphism located in the 72nd codon, inducing a tube-like structure involved in the activation of angiogenesis and an increase in the expression of some differentiation clusters and chemokines. Other studies have shown that some mutations can affect the 47th codon in the *TP53* gene, associated with a decrease in the efficacy of apoptosis, which could account for severe damage caused to tumor-suppressing functions, the gain of some oncogenic features, and genome instability combined with a lack of cell cycle control [44,47].

Furthermore, the mutated form of p53(mtp53) is commonly noted in ER-negative breast cancer, generated by missense substitution mutations, specifically G to C conversions; these mutations have been found in 43% of non-basal TNBC and 62% of basal-like cases [47,50]. This latter phenotype has the worst overall survival rate and a high risk of metastasis. However, patients with wtp53 associated with ER-negative breast cancer have a better prognosis, while wtp53 associated with ER-positive breast cancer has a poor prognosis [51,52]. 

#### 2.2.2. BCL2

BCL2 (B-cell lymphoma-2) is a mitochondrial integral membrane protein acting as an antiapoptotic, promoting pro-oncogenic and/or oncogenic pathways; it is correlated with cell growth and proliferation and promotion of genomic instability [53]. The utility of BCL2 as a biomarker in TNBC is yet to be demonstrated, since BCL2 expression is regulated by estrogens and is commonly detected in ER-positive BC. In particular, this trait is prevalent in hormone-receptor-positive, node-negative breast cancers and could be used as a prognostic marker [54,55]. According to several studies, the absence of BCL2 has been associated with a better response to neoadjuvant chemotherapy based on doxorubicin. In another context, patients expressing BCL2 deficiency treated with anthracycline had a better outcome than those receiving 5-FU, cyclophosphamide, or declined chemotherapy [56].

### 2.3. Regulators of DNA Damage Response

#### 2.3.1. BRCA1/2 and PARP Inhibitors

The *BRCA1* and *BRCA2* genes are located on the 17th and 13th human chromosomes, respectively [57]. These entities have multiple functions and influence several DNA repair mechanisms, such as the recruitment of DNA repair proteins, activation of some checkpoints, and transcriptional regulation of DNA damage. They can also control cell division, cellular proliferation, and differentiation [58]. *BRCA1* mutations are frequently the main driver of germ-line mutations (gBRCA), increasing breast cancer risk by 50% to 85%, especially in the triple-negative form [59]. Some studies have found that the majority (80%) of breast cancer cases expressing the *BRCA1* hereditary mutated form can be classified as triple-negative breast cancers (TNBC) and/or basal-like breast cancers (BLBC) [60], of which TNBCs represent 15% to 20%. Furthermore, the overall survival of patients expressing gBRCA exceeds that of patients free of *BRCA* mutation. Nevertheless, the BRCA phenotype includes patients displaying *BRCA1/2* sporadic mutations, such as epigenetic inactivation associated with DNA damage repair mechanism deficiency, particularly homologous recombination defects [61]. 

According to the national center of biotechnology information (NCBI), BRCA1/2 proteins combine with other suppressor proteins to form a large-scale multi-functional complex known as the BRCA1-associated genome surveillance complex (BASC). As a result, several DNA damage types, such as double-stranded breaks (DSB), undergo an active repairing process through two different types of defense mechanisms: homologous recombination (HR), an error-free machinery ensuring genomic stability, and non-homologous end joining (NHEJ), which is an inaccurate operating repair mechanism that may deepen the DNA damage response (DDR) [62]. Under physiological conditions, HR mechanisms require the presence of multiple key proteins, e.g., BCRA1/2, RAD51, PALB2, etc., and a DSB phenotype to regain the primary form of the DNA. Therefore, some sporadic mutations can be caused by various alkylating agents (ex: platinum) and the outcome of PARP (poly-ADP ribose polymerase) inhibitors [63]. 

Loss of BRCA function makes DNA sensitive to some damaging agents. However, cells have numerous repair mechanisms, including the PARP pathway, to maintain their stability. Recently, PARP inhibitors have been used as a treatment in chemotherapy; these agents can trap the PARP1 proteins as their ligands at the single-strand break (SSB) sites, which will lead to blocking of the reading frame and stopping of the replication [64]. This arrest will collapse the reading frame and create a DSB, which will appeal to the HR mechanism. This latter is defective in the BRCAness phenotype, exposing the tumor cells to therapeutic vulnerabilities and maintaining a synthetic lethality, leading to cell arrest and apoptosis due to the absence of high-fidelity repair mechanisms (NHEJ) [65]. 

#### 2.3.2. Topoisomerase

Topoisomerases are DNA ligase enzymes; their main function is to maintain the topographic structure of DNA by reducing the tension in the DNA’s double helix. These enzymes are present to reduce the blockage of advancing replication forks, which can lead to chromatin rupture, replication error, and, eventually, cell arrest. They can be divided into topoisomerase I and topoisomerase II, of which the complex DNA-TOP1 prevents the DNA legation of DSB, which can ensure time for repair mechanisms to repair the errors or lead the cell to apoptosis in case of the absence of DNA repair [66]. Topoisomerase II is highly concentrated in the cell division phase, since it is associated with cell proliferation [67]. 

Topoisomerase inhibitors are enzymes that control the topographic structure of DNA by inducing transient cuts that will be overtaken later without correction so the two DNA ends do not re-join; these will be occupied by the enzyme, inducing cell cycle arrest in G2. Topo II inhibitors can be used as drugs against breast cancer cells by activating the p38 gamma-related protein (a key protein belonging to the MAP kinase family), implicated in several major mechanisms for the transduction of extracellular signals. As a result, this can lead to the phosphorylation of Topo II, making it a straightforward target inhibiting proliferation [68].

#### 2.3.3. PTEN

PI3K/AKT is the most investigated oncogenetic signaling pathway. The relevant alterations accompanying this are associated with poorer prognostic and treatment resistance in TNBC [69]. This common pathway includes multiple proteins, such as PIK3Ca, AKT, and PTEN. Alterations leading to the loss of PTEN induce both excessive cell division and proliferation and inhibit apoptosis [70,71]. This is because PTEN is a key tumor suppressor through the phosphorylation/dephosphorylation mechanisms of PIP3 (phosphatidyl-inositol triphosphate), controlling the inhibition of AKT, resulting in paralysis of the PI3K/AKT pathway [72,73,74]. 

The loss of PTEN is related to the genomic or proteomic level (frameshift mutations, or the production of a truncated protein), which causes the hyperactivation of AKT, a phenotype expressed in 15% of TNBC patients and which could be used as a biomarker [70,75]. 

The initial stages of tumor development are conducted through a proficient group of proteins, such as P53, AKT, PIK3CA, and PTEN. The cancer genome atlas network expresses that this union is the basic mutation niche and is the most-targeted pathway in multiple-therapy studies. PTEN is known as the stabilizer of P53 through the association between the two units, preventing the latter from achieving ubiquitination [73]. Despite this critical role, the loss of PTEN, accompanied by the activation of PI3K, produces an increase in P53 expression in TNBC [76]. In summary, targeting only one segment in this series of interconnected proteins leads to modest overall survival, suggesting that a complete exploration of the global pathway could lead to a better outcome than a single selection [76,77]. 

### 2.4. Immune Checkpoints

#### 2.4.1. PD-1/PD-L1

The immunosuppressive phenotype expressed by breast cancer cells has been described by several studies; this tumor microenvironment shields these cells from the immune system. The transmembrane receptor programmed cell death protein 1 (PD-1) encoded by the *PDCD1* gene is expressed in a variety of cells, including tumor cells and tumor infiltration immune cells, such as antigen-presenting cells (APC) and T lymphocytes. It is activated by the PD-L1 (programmed cell death protein ligand 1) protein and PD-L2 (programmed cell death protein ligand 2). The expression of these receptors, predominantly in tumor cells, ensures interaction with the immune system, self-tolerance of the tumor expansion. Thus, downregulation of T lymphocytes and their associated cytotoxicity through the production of cytokines such as IFN-γ is a major factor in the proinflammatory mechanisms implicated in increased cancer development [78].

PDL1 is expressed in 20% of TNBCs and is characterized as the most immunogenic against the other sub-types; it is associated with the poorest prognosis in younger generations and is implicated in higher grade, larger tumor size, and ER-negative status. However, PDL1 expression has been proven to improve pCR rate, overall survival, and metastasis-free survival [79]. Several studies have shown that the combination of antibodies directed against tumor cells and different well-known therapies prove the possibility of a revolutionary age in cancer targeting. Some preclinical studies have expressed the possibility of combining PD-L1 inhibitors and BRAF/MEK inhibitors, which have demonstrated promising results in other solid tumors, such as neck and head carcinomas [80,81]. Mittendorf and her associates suggest that an analog of MAPK affecting the PI3K/AKT/ mTOR pathway has proven a downregulation of PD-L1 expression connected to PTEN deficiency. It is known that activation of the immune system through the conversion of the immunosuppressive phenotype to an immunostimulatory type through the arousal of CD4+ and CD8+ T cells reduces tumor charge and increases regulatory T lymphocytes [82,83,84]. 

PD-1 and PD-L1 are implicated in the bio-identification of triple-negative breast cancer, highlighting an emerging field to investigate in immunotherapy as a new approach for therapeutic purposes. Furthermore, it is critical to explore several immune pathways and restore T-cell function by restraining the damage driven by tumor cells. 

#### 2.4.2. Mismatch Repair Mechanism Deficiency (dMMR)

To address the involvement of DNA repair mechanisms in DNA damage response (DDR), several researchers have investigated the accuracy of the mismatch repair mechanism in multiple solid tumors, such as breast cancers. This mechanism is considered a highly conserved damage repair mechanism due to its fundamental approach to preserving genome integrity. MMR has been identified as the pathway aiming at base substitution, serving to restore the physiological state damaged by DNA replication errors [85]. The loss of this mechanism has been associated with microsatellite instability (MSI) (i.e., a genomic condition of hypermutability with a phenotype related to the loss of MMR) [86]. It has been identified in different studies that loss of MMR could be associated with an elevated tumor immunological burden score, leading to apoptosis evasion and the development of malignant mutations. 

The MSI-high phenotype (microsatellite instability) is expressed in multiple germline cancers and some dominant sporadic mutations linked to the development of a large spectrum of solid tumors [87,88]. In the TNBC context, studies have estimated that the frequency of MSI-H dMMR is less than 2%, which is minor relative to highly expressed biomarkers. Ren et al., based on 440 cases, compiled TNBCs and investigated them through multiple approaches, such as immunohistochemistry performed on FFPE blocs, pentaplex PCR, and next-generation sequencing (NGS); MMR protein expression analysis revealed mostly proficient MMR (MMR). Even though 92.8% of the cases were of MSS (microsatellite stability), one case revealed the loss of MSH-2 and was confirmed as MSI-low by pentaplex PCR [89]. 

The dMMR/MSI aspect is related to prognosis and immunotherapy response; however, potential targeted therapies differ between tumor types. dMMR/MSI-H has been associated with poor prognosis in patients expressing resistance to 5-fluorouracil (FU)-based adjuvant chemotherapy in colorectal cancer [90], while dMMR/MSI-H was associated with good prognosis in gastric cancer [91]. According to Wild and his team, it has been shown that dMMR/MSI-H correlates with resistance to chemotherapy and poor prognosis [92], while others have reported that patients with dMMR/MSI-H have lower overall survival than ER-negative breast cancer patients treated with chemotherapy [93]. 

#### 2.4.3. Notch Pathway

The notch family consists of four receptors (Notch1, Notch2, Notch3, and Notch4) interacting with five ligands (DL1, DL3, DL4, Jagged 1, and Jagged 2). This signaling pathway has been implicated in pleiotropic effects, assisting the development process of several tissues [94]. It has been proven that Notch1, and its ligand, Jagged 1, are expressed in human breast cancer, particularly in basal-like mesenchymal stem-like TNBC subtypes. These latter are distinguished by a dysregulated Notch signaling and over-expression of NOTCH1 [95,96]. Moreover, some studies have demonstrated that C-terminal deletion of NOTCH1 intracellular domain (N1ICD) increases its instability without any effect on the amplification and recapitulation of N1ICD-dependent signaling; this contributes to the development of breast cancer [97]. In TNBC, gain of function in *NOTCH 1* occurs in 10% of proven cases [98]; however, around 13% of TNBC cases expressed life extension of N1ICD-dependent signaling or receptor activation independent of the attachment of its ligand due to in-frame deletion of the *Notch1* gene, particularly in exons 21 and 27 [99]. Accordingly, multiple Notch-associated genes have been affected, such as NOTCH 3, HES1, and MYC. 

Zhu and his colleagues identified a significant correlation between activation of the Notch pathway and stimulation of the AKT and NF-kB pathways [100]. Additionally, the expression of NOTCH1 and its ligand JAGGED1 is implicated in angiogenesis observed in the adjacent endothelial cells, which is associated with poor overall survival and could be related to bone metastasis [101]. However, the expression of mRNA is linked to recurrent TNBC [101,102]. The active role of the Notch pathway in several physiological mechanisms, such as proliferation, apoptosis, and the maintenance of stem cells recorded in the tumor immune microenvironment has led to an effective approach in studies dedicated to the investigation of new inhibitory drugs accommodating this pathway; notably, initial studies have declared a remarkable effect in elevating the sensitivity of patients with TNBC to chemotherapy [103]. 

### 2.5. Epigenetic Modifications

#### 2.5.1. DNA Methylation

DNA methylation is a reversible mechanism consisting of adding a methyl group to the fifth carbon atoms of cytosine bases in CpG-rich promoter sequences. Inversely, the demethylation reaction consists of removing the methyl group from the cytosines, which is catalyzed by ten–eleven translocation (TET) demethylase proteins (TET1, TET2 and TET3). DNA methylation is a normal epigenetic process contributing to the regulation of gene function. However, a large number of studies highlight the involvement of DNA hyper- or hypomethylation in breast tumorigenesis [104,105]. In fact, TNBC tumors present genome-wide hypomethylation, correlated with worse overall survival (OS). TET1 DNA demethylase is specifically overexpressed in about 40% of TNBC patients. Bioinformatic analyses of breast cancer cell line panels reveal a network linking TET1 to the hypomethylation and activation of several oncogenic pathways, such as PI3K, EGFR, and PDGF [106]. Furthermore, hypomethylation of oncogenes like *ADAM12* or lncRNA HUMT causes their overexpression in TNBCs. Mendaza et al. demonstrated that the *ADAM12* gene (disintegrin and metalloproteinase domain-containing protein 12) was exclusively hypomethylated in TNBC, which was associated with a worse outcome. *ADAM12* silencing attenuated TNBC cell proliferation and migration and improved doxorubicin sensitivity [107]. Moreover, long non-coding RNA HUMT hypomethylation promotes lymphangiogenesis and metastasis by activating FOXK1 in TNBC tissues [108]. These findings suggest ADAM12 and lncRNA HUMT as potential therapeutic targets in TNBC. 

On the other hand, a hypermethylation phenotype in specific genes has also been identified in TNBC tissues despite global hypomethylation [109]. DNA methylation is catalyzed by the DNA cytosine methyltransferase family members, including DNMT1, DNMT3A, and DNMT3B. DNMT1 is mostly overexpressed in breast cancer, especially in TNBC. DNMT3A and DNMT3B are also reported to be overexpressed in TNBC tissues [105,110]. Methylation of key genes can result in gene expression silencing and, consequently, contribute to tumor cell spread and metastasis. This applies to the hypermethylation of CpG promoters of genes involving in DNA damage response (BRCA1, MLH1, MGMT), tumor suppressor genes (RASSF1A, CDKN2A, CDKN1B, CCND2a), apoptosis (HOXA5, TMS1), adhesion, invasion (TWIST1, CDH1, TIMP3), and cellular detoxification genes (GSTP1) [24].

#### 2.5.2. Long Non-Coding RNAs (lncRNAs)

Non-coding RNAs (ncRNAs) can be classified into two groups depending on transcript size. Small ncRNAs are usually shorter than 200 nucleotides. They include microRNAs, piwi-interacting RNAs, small nucleolar RNAs, and short interfering RNAs. The second group comprises long non-coding RNAs with lengths exceeding 200 nucleotides. It includes intergenic lncRNA, intronic lncRNA, sense lncRNAs, antisense lncRNAs, enhancer lncRNAs, and bidirectional lncRNA [111].

Several studies have identified the role of lncRNAs in different biological processes, including apoptosis, cell proliferation, cell differentiation, metastasis, and chromatin modification. Abnormal expression of different lncRNAs has been identified in a large number of tumors, including TNBC [112].

LncRNAs can bind miRNA, compete with miRNA-targeted mRNAs, and, consequently, affect miRNA-mediated gene regulation. They constitute a post-transcriptional regulatory network involving mRNAs and lncRNAs, named the competing endogenous RNAs network (ceRNA). This is a significant pathway for lncRNA-modulated post-transcriptional regulation in TNBC [113]. From this perspective, Le et al. used weighted gene co-expression network analysis besides the existing microarray mRNA and lncRNA expression data of TNBC. They established a ceRNA network and found that two mRNAs (RAD51AP1 and TYMS) were correlated with overall survival. They concluded that TNBC-specific mRNA and lncRNAs could contribute to a ceRNA network, presenting a possible therapeutic target for the treatment of TNBC patients [114]. At the same time, Song et al. screened specific lncRNAs, miRNAs and mRNAs in TNBC from the cancer genome atlas database and identified five molecules (TERT, TRIML2, PHBP4, mir-1-3p, mir-133a-3p) associated with the prognosis of TNBC with a prognostic ceRNA sub-network between those molecules. qPCR was used in order to validate the results and revealed a high expression of the *TRIML2* gene in TNBC tissues and cells compared with normal paracancerous tissues and mammary cells [115].

It has been increasingly recognized that lncRNAs can play an important role in the regulation of cell proliferation and apoptosis. For instance, lncRNA GAS5, through miR-196a-5p and miR-378a5p/SUFU signaling, was identified as an apoptosis promoter and proliferation inhibiter of cancer cells in patients with TNBC [108,116]. Furthermore, lncRNA HEIH was reported to regulate cell proliferation and apoptosis in TNBC by targeting the miR-4458/SOCS1 axis [117]. LncRNAs might also interact with transcription factors or histone-modifying enzymes and regulate the expression of their target genes. Indeed, LINC02095 was identified as promotin the expression of SOX9, the oncogenic transcription factor, leading to cancer cell proliferation [118]. Wu et al. reported that LINC00152 might increase tumorigenesis in TNBC patients via inactivation of the BRCA1/PTEN by DNA methyltransferases [119]. Some lncRNAs involved in the cell proliferation and apoptosis process in TNBCs can also be regulated by other LncRNAs. In fact, Yu et al. demonstrated that lncRNA DRHC inhibited the proliferation of TNBC cell lines by downregulating lncRNA HOTAIR [118]. Nevertheless, HOTAIR overexpression did not affect the expression level of lncRNA DRHC [120]. 

LncRNAs can also bind to specific DNA sequences and consequently regulate gene expression. Wang SW et al. have demonstrated that lncRNA MIR100HG binds to the *p27* gene and leads to the formation of a complex RNA–DNA structure, regulating the expression of p27 and consequently influencing cell proliferation in TNBC [121].

Numerous studies have reported that lncRNAs may be involved in tumor invasiveness and metastasis. Han et al. found that lncRNA CCAT1 promotes TNBC progression by targeting the miR-218/ZFX axis [122]. Furthermore, Zhang et al. demonstrated that the up-regulation of NAMPT-AS (a long noncoding antisense RNA of the nicotinamide phosphoribosyltransferase gene) is associated with metastasis and poor prognosis in TNBC. Consequently, NAMPT-AS/NAMPT could be a promising therapeutic target in patients with TNBC [123].

Moreover, lncRNAs could be implicated in the regulation of drug resistance in TNBC. LncRNA GAS5 is one of the most studied lncRNAs involved in this regulation [124]. In fact, in TNBC patients, the expression of GAS5 is strongly associated with tumor resistance to several chemotherapeutic drugs, such as cisplatin, adriamycin, and paclitaxel [125]. Furthermore, GAS5 expression might decrease sensitivity to mTORC1 inhibitor rapalogues, as well as to the dual mTORC1/mTORC2 inhibitor AZD8055 [126]. In addition, lncRNA LINP1is known to activate DNA double-strand break repair through the NHEJ pathway, leading to ionizing radiation resistance. Blocking LINP1 might help enhance the sensitivity of the TNBC tumor to radiotherapy [127]. Hence, several studies have highlighted the possibility of altering lncRNA expression in order to avoid drug and/or radiotherapy resistance in TNBC.

More recently, a novel long noncoding RNA Uc003xsl.1 was found to be upregulated in human TNBC tissues. In fact, Uc003xsl.1 was demonstrated to be directly bound to nuclear transcriptional factor NF-kB-repressing factor (NKRF), causing the activation of the NF-kB/IL8 axis. Therefore, lncRNA Uc003xsl.1 acts as an oncogene and strongly enhances TNBC progression. Targeting Uc003xsl.1 in order to suppress the NF-kB/IL8 axis might represent an interesting therapeutic mode in TNBC treatment [128].

#### 2.5.3. MicroRNAs (miRNAs)

Several studies have demonstrated an association between many miRNAs and disease progression mechanisms in TNBC, including epithelial-to-mesenchymal transition (EMT), migration, invasion, and metastasis [129,130]. 

In fact, EMT is considered a crucial process in which cancer cells acquire migratory and invasive properties leading to tumor metastasis [131]. Many studies indicate the involvement of miRNAs in the mechanism of EMT. For instance, the miRNA-200 family, comprising miR-200a, miR200b, miR-200c, miR-141, and miR-429, is believed to include the predominant miRNAs acting as negative regulators of EMT. MiR-200a is an inhibitor of EMT, preserving the epithelial phenotype through the targeting of E-cadherin repressors ZEB1 and ZEB2 [132]. Moreover, miR-200b restrains TNBC migration and metastasis through various mechanisms involving inactivation of EGFR and the downstream PI3K/Akt signaling pathway and suppresses FUT4 expression in cultured breast cancer cell lines [133]. Meanwhile, the epithelial phenotype is conserved by inhibiting the genes of cell motility, suppressing cell proliferation and promoting apoptosis in TNBC [134]. For all these reasons, the miR200 family is currently considered a potential therapeutic target in the management of TNBC [135].

On another side, many miRNAs are involved in migration, invasion, and metastasis. For instance, Liu et al. revealed that the overexpression of lncRNA AWPPH and miR-21 promotes cancer cell proliferation [136]. Moreover, the combination of miR-21 with the 3’ UTR of LZTFL1 (leucine zipper transcription factor-like 1) activates cell proliferation and metastasis [137]. Fang et al. demonstrated that miR-21 was upregulated, promoting the proliferation and invasion of MDA-MB-468 cells, in TNBC. Meanwhile, PTEN was downregulated by miR-21 in these same cell lines. This shows the importance of considering the inhibition of miR-21 and the overexpression of PTEN as a potential strategy in the diagnosis and prognosis of patients with TNBC [138].

To summarize, Table 2 represents a group of selected microRNAs associated with TNBC biology and their validated targets and functions. 

#### 2.5.4. Short Interfering RNA

Numerous techniques are employed to selectively silence genes implicated in cancer physiopathology. RNA interference is one of these novel methods for controlling gene expression. In fact, in this system, short double-stranded RNAs (dsRNAs) as well as microRNAs (miRNAs) can be transformed into short interfering RNAs (siRNAs), whose main role is to silence one or more specific genes through hybridization with its messenger RNA target [158]. Recently, with the advancement of nanotechnology, nanoparticles have become a promising approach for encapsulating and transporting siRNA with high efficiency along with low toxicity [159].

Hamurcu et al. have demonstrated that FOXM1 is strongly upregulated in TNBC. The knockdown of FOXM1 by RNA interference considerably inhibits eEF2K expression causing suppression of cell proliferation, migration and invasion and induction of apoptotic cell death. Therefore, this study highlights the potential of the FOXM1/eEF2K axis as a molecular target in TNBC and other types of cancers [160].

In another study, researchers found that the *POLR2A* gene is frequently co-deleted with *TP53*, and a large number of TNBCs harbor these deletions (53%). They used siPol2-laden nanoparticles (siPol2@NPs) to precisely suppress the expression of *POLR2A,* contributing to the enhanced growth reduction of tumors. The results reveal the potential of targeting POLR2A as a therapeutic strategy for TNBC [161].

Parvani et al. used β3 integrin as a therapeutic target to treat TNBC by delivering β3 integrin siRNA through lipid ECO-based nanoparticles. This treatment can successfully silence β3 integrin expression, attenuate EMT and invasion, and restore cytostasis [162]. 

Currently, several siRNA-mediated nanoparticles have entered phase I and II clinical trials for the treatment of various cancers, including TNBC. For instance, EphA2-targeting DOPC-encapsulated siRNA against ephrin type-A receptor 2, also referred to as EPHARNA, was studied in 40 patients presenting spreadable solid tumors [163]. In vitro and in vivo research has revealed that EPHARNA has anti-angiogenic effects including important reductions in tumor growth. Furthermore, the concomitant administration of paclitaxel and EphA2-siRNA-DOPC inhibited tumor growth. Safety studies with EPHARNA were conducted with success in rhesus monkeys, and it was well tolerated at all doses tested. These promising results have led to an ongoing first-in-human Phase I clinical trial (NCT01591356) [163].

Table 3 summarizes the TNBC biomarkers.

## 3. Omics Approaches to Exploring TNBC

Over the past decade, many attempts have been made to unravel the molecular mechanisms of carcinogenesis using omics approaches, such as scanning the genome for cancer-specific mutations, identifying altered epigenetic landscapes in cancer cells, and exploring the differential expressions of mRNA and proteins through genomics, epigenomics, and transcriptomics techniques, respectively [164]. 

Although they have not been completely turned into routine methods, omics approaches are widely used in research and allow the identification of gene groups (genomic signatures), patient stratification through assessment of disease progression, and understanding of possible responses to treatment [165]. 

### 3.1. Genomic

In oncology, genomics approaches have focused on DNA sequencing to identify cancer-specific mutations and analyze chromosomal rearrangements. DNA isolated from cancer cells carries a wide range of somatic and potentially heritable changes, ranging from single-nucleotide polymorphisms, insertions, and deletions to copy number variations and larger structural rearrangements [166,167]. 

Early studies in TNBC were interested in identifying inherited mutations that significantly increase cancer risk. A significant number of these variants are now routinely tested as adjunct diagnostics for tumor characterization and treatment decision-making.

Other studies have focused on the analysis of chromosomal rearrangements and copy number variations in cancer cells. Numerous technologies are used, including fluorescence in situ hybridization, comparative genomic hybridization, and, more recently, whole-genome sequencing, which is also used to characterize gene conversion by comparing tumor tissues from multiple cancers with adjacent tumor tissues [168]. 

Some of these genomic signatures have been validated in large cohorts and have been the precedent for commercial genomic tests, providing complementary information to clinicians to obtain more precise classification of a patient’s cancer where it has a high risk of recurrence, in order to offer more personalized management [169]. 

### 3.2. Transcriptomic

Transcriptomics examines RNA levels, both qualitatively (transcripts presenting at a given time, identification of splice sites) and quantitatively (the quantity expressed of each transcript). It provides a link between the genome, the proteome, and the cellular phenotype.

RNA sequencing studies have identified thousands of new isoforms and have shown the transcriptome’s complexity [170]. An even more important contribution of these studies has been the development of the field of non-coding RNA.

In the early 2000s, transcriptomic analyses of TNBCs were carried out on a large number of genes potentially involved in oncogenesis, control of cell growth, cell differentiation, and cell death.

Molecular taxonomies established by transcriptomic studies are important for characterizing alterations in the genetic programs of tumors and their complexity [171]. However, they have several limitations that do not make them, at the moment, directly applicable routinely at the clinical level.

Widely distributed and commercially available assays include MammaPrint^®^ (Agilent, Amsterdam, The Netherlands), the Oncotype DX^®^ test (Genomic Health, Redwood City, CA, USA), and the PAM50 RNA-based gene signature [169].

### 3.3. Epigenomic

Epigenomics is defined by the genome-wide identification of chemical modifications such as methylation and acetylation of DNA and/or DNA-binding histone proteins, which serve as a major regulatory mechanism controlling gene expression and cellular phenotypes [172]. 

Recent studies indicate that matrix alterations in TNBC cells are epigenetically regulated [173].

Epigenomics techniques that are widely used include chromatin immunoprecipitation (ChIP) assays coupled with next-generation sequencing and targeted and genomic bisulfite sequencing. Using these approaches for targeting tumor microenvironment and epigenetic pathways, alone or in combination, represents a potential therapeutic strategy in TNBC [173]. 

### 3.4. Proteomic

Proteomics is a new branch of biochemistry that aims to study proteins on a large scale.

The rapid advancement of proteomics offers a unique opportunity to quantitatively profile the activation states of actionable key signaling intermediates and phosphoproteins in TNBC samples. 

There are two types of targeted proteomics approaches: non-MS-based methods that use protein detection antibodies (Western blots, ELISA, immunohistochemistry, and reverse-phase protein arrays) and MS-based methods (mass spectrometry imaging, targeted proteomics, and next-generation proteomics) [174].

### 3.5. Metabolomic

Metabolomics is the analysis of small molecules (called metabolites) present in a cell, tissue, or fluid that reflect changes in the functions of mediating enzymes and proteins.

Advances in liquid chromatography–tandem mass spectrometry (LCMS/MS) techniques for precise quantification and data analysis have made it a solid platform for the simultaneous identification and quantification of metabolites.

This makes it possible to study the states of activation of many metabolic pathways through statistical analysis and by comparison with databases. Metabolomics data have the advantage of reflecting the last steps in the different cascades of signals exerted on tissues [175]. 

This approach has already provided relevant results in the context of breast cancer. However, systematic characterizations of metabolites in triple-negative breast cancer (TNBC) are still lacking [176]. 

## 4. Updated Neoadjuvant Treatments for TNBC

Despite the absence of ER, PR and HER2 expression in the TNBC group, neoadjuvant treatment has shown strong efficacy [177]. This is based on anthracycline and taxane, which have been used to decrease the sizes of tumors and to perform surgery on non-operable tumors. Currently, neoadjuvant therapy is used in testing novel therapies and response to treatment is linked to the outcomes in patient groups.

Treatment with neoadjuvant chemotherapy (NACT) showed a substantial benefit in TNBC. Indeed, retrospective analyses of several clinical trials have concluded that a lack of ER expression predicts a better response to cytotoxic chemotherapy: overall survival (OS) was 76% for patients with TNBC receiving high-dose chemotherapy, compared with 61% in the dose-dense arm (hazard ratio = 0.58; confidence interval = 95%) [178]. Additionally, other studies [179] have revealed a strong correlation between pathological complete response (pCR) and an increased number of cycles of chemotherapy. Nevertheless, it is crucial to specify that patients with TNBC tumors may respond to NACT rapidly, shortly after two cycles of therapy [180]. Other authors [181,182] have shown a large benefit with dose-dense or dose-intensive chemotherapy in TNBC tumors. 

These neoadjuvant treatments improve, firstly, pCR rates and, secondly, disease-free survival (DFS) [183]; for this reason, pCR is considered a predictive marker for long-term clinical response and survival in TNBC.

### 4.1. Treatment with Platinum in TNBC

The combined use of two drugs (anthracyclines and taxane) in patients affected with TNBC has become current standard clinical practice. Several studies [184] have shown that treatment with a higher dose of anthracycline (≥300 mg/m^2^) combined with taxane (≥400 mg/m^2^) achieves higher pCR rates; HR = 1.49 and 1.73 respectively. However, pCR response increased significantly (10–15%) when we combined platinum with taxane (four cycles of cisplatin at 75 mg/m^2^ every 21 days) [185,186,187]. Platinum agents are DNA-damaging, with alkylating cytotoxic activity in TNBC groups [188,189]. They exert a synergic effect when added to taxane-based neoadjuvant chemotherapy treatment [190]. A combination of platinum with anthracycline/taxane-based NACT showed a large benefit on pCR rate and DFS/OS [191]. Although other agents combined with standard NACT produce no improvements in TNBC outcomes [192,193], in older patients with comorbidities or cardiac risk non-anthracycline regimens may be proposed. A true benefit was observed with platinum; nevertheless, the percentage of recurrence was higher than in the other breast cancer groups; thus, the development of new biomarkers or targeted agents could be crucial to improving treatment strategy in NACT and the prognosis of TNBC patients by increasing pCR rates.

Large TNBC tumors were associated with *BRCA1-2* mutation, which showed a higher pCR than wild-type TNBCs [186,194]. The last type increased the pCR rate when we added carboplatin to NACT [195]. It seems that the pCR was linked with homologous recombined deficiency HRD score [196].

### 4.2. Role of Immunotherapy in the Treatment TNBC

Currently, immune oncology (IO) is a novel approach to treating tumors by activating the immune system [197]. Immune checkpoint inhibitors have shown success against malignant cells and a successful response [197,198]. 

TNBC tumors contain higher levels of T regulator lymphocytes, PD-1-positive lymphocytes, and PD-L1-positive myeloid cells compared with luminal tumors [193]. Otherwise, these tumors are characterized by high tumor mutational burden (TMB) and high levels of intra-tumoral tumor-infiltrating lymphocytes (TILS), which confer a good response to immunotherapy in TNBC patients [199,200], who experienced better clinical benefit than other breast cancer groups [201]. Several immune checkpoint biomarkers, including PD-L1 inhibitors (atezolizumab, durvalumab) and PD1 inhibitors (pembrolizumab, nivolumab, and camrelizumab) have recently been used in monotherapy and co-therapy for TNBC in trials and have shown improvements in survival outcomes when combined with chemotherapy [202,203,204]. Indeed, pembrolizumab and atezolizumab have been approved for advanced-stage PD-L1 positive TNBC. The emergence of these biomarkers will be key in personalized immunotherapy in future TNBC treatment [205].

### 4.3. PD-L1/PD1 Therapy

Several studies have been conducted combining immunotherapy agents with NACT in TNBC tumors. Antibodies targeting PDL1, atezolizumab and durvalumab, have a beneficial impact on pCR and immune system activity. This is why, recently, the FDA approved the addition of atezolizumab to Abraxane (nab-paclitaxel) for unresectable TNBCs expressing PDL1 [206], because survival (OS) was significantly improved for these patients, particularly in the early stage of TNBC [206]. Furthermore, we noted a higher pCR and better safety in TNBC patients receiving neoadjuvant therapy (azolizumab coupled with nab-paclitaxel and anthracycline-based treatment) [207]. Meanwhile, the addition of durvalumab increased, but not significantly, pCR rates in all patients [208]. Other authors suggest linking the expression level of PDL1 and the increase in pCR in BC patients receiving durvalumab with NACT [209].

In July 2021, a PD1 inhibitor (pembrolizumab) received FDA approval for the treatment of TNBC; this is used for early-stage high-risk TNBC in combination with NACT as a neoadjuvant followed by a single agent as an adjuvant treatment after surgery. 

Pembrolizumab combined with NACT produced an increase in the pCR rate that ranged from 49% to 71% and few side effects [210]. This association between chemotherapy and PD1 blockers had a real positive impact on antitumor activity and a significant objective response rate (ORR) [203]. In the same direction, pembrolizumab combined with NACT improved 3-year EFS (event-free survival) from 76.8% to 84.5% (HR 0.63, *p* = 0.0003) [211], and this was specially improved in invasive residual disease (from 56.8% to 67.4%) and pCR subgroups (from 92.5% to 94.4%) [212]. Moreover, therapy with a combination of camerelizumab (anti PD1) and apatinib (anti-VEGF) significantly enhanced ORR. This was a better therapeutic approach in advanced TNBC than camerelizumab or apatinib alone [213]. IO biomarkers associated with conventional therapy produced a large improvement in patient outcomes in TNBC; nevertheless, it is necessary to manage the toxicities of these agents [214]. 

### 4.4. Antibody-Drug (ADCs) Conjugates

ADCs are human monoclonal antibodies and include a variety of molecules. They are currently considered promising therapeutic agents for tumors, especially for TNBC. These antibodies have demonstrated high efficacy in large human cancers by targeting a specific antigen of the tumor and unloading cytotoxic drugs.

Trastuzumab deruxtecan (T-DXD) is a conjugate of Her2 and a DNA topoisomerase I inhibitor. It has shown persistent antitumor activity and has significantly increased OS. Overall, 94.1% of patients with HER2-positive metastatic breast cancer treated with T-DXD survived 12 months later [215]. ADC drugs will soon be used in clinical trials as future therapeutic agents against TNBC tumors [203].

### 4.5. Novel and Targeted Agents

#### 4.5.1. PARP Inhibitors

TNBC tumors have more *BRCA* mutations than other BC subtypes (20% versus 5%). PARP inhibitors are considered an antitumor therapy for these patients [216].

The main function of the PARP enzyme is to maintain genome stability and participate in cell progression and apoptosis. PARP inhibition leads to loss of the DNA repair function and cell apoptosis through p53 activity. Thus, PARP inhibition is the best target in TNBC tumors with a *BRCA1/2* mutation (BRCAm). Benefits were obtained using olaparib and talazoparib (PARP inhibitors) in monotherapy compared with chemotherapy in germline BRCA-mutated metastatic breast cancer patients [217,218]. However, tumors without *BRCA* mutations receive no benefit from PARP inhibitors in treatment [219]. Based on several positive studies, olaparib was approved for the treatment of BRCA-mutated BC in January 2018. Indeed, olaparib improved progression-free survival compared with chemotherapy in patients with germline *BRCA1/2* mutations and Her2-negative metastatic breast cancer [220]. 

Study of the combination of PARP inhibitors with NACT is also ongoing to evaluate their possible synergic effect. Additionally, the combination of a PARP inhibitor/immune checkpoint inhibitor and NACT is also of great interest since it has improved PCR in TNBC from 27% to 47% [221].

#### 4.5.2. Androgen Receptor (AR)

The positive androgen receptor is expressed in 10–15% of TNBC patients [222], and a promising effect has been shown in advanced AR+ TNBC patients treated with abiraterone, bicalutamide, and enzalutamide [223,224]. This is still to be confirmed in early-stage TNBC. 

#### 4.5.3. Growth Factor Receptor (EGFR, VEGFR, and FGFR) Inhibitors

EGFR is often overexpressed in TNBC tumors and could be a promising treatment target. However, it has been reported that the addition of cetuximab to carboplatin increased progression-free survival and overall survival insignificantly [225]. Studies suggest that EGFR activity is ligand-independent, and did not observe a benefit of EGFR inhibition in an unselected population with TNBC. It may be a negative prognostic factor in these tumors [226].

Vascular endothelial growth factor (VEGFR) is implicated in breast cancer angiogenesis [224], but inhibition with VEGFR kinase increases response in TNBC. Bevacizumab, a monoclonal antibody against VEGFA, increases pCR in patients with TNBC when they are treated with epirubicin, cyclophosphamide, and docetaxel, from 27.9 to 39.3%, *p* = 0.003 [224]. In addition, patients with *BRCA1* or *BRCA2* mutation treated with bevacizumab showed significantly higher pCR rates [227]. Amplifications or mutations in *FGFR* are very rare in TNBC. These tumors could display clinical benefits with targeted therapies.

Figure 1 summarizes the signaling pathways as well as the different therapeutic targets in tumor cells in triple negative breast cancer.

## 5. Conclusions

Previously considered an unapproachable disease by molecular therapy, TNBC has recently been the focus of fruitful investigation for the incorporation of new targeted therapies. Such advances are due to progress made in understanding the molecular biology of TNBC as well as technological advances in diagnosis and monitoring. It is with great optimism that we look forward to current and future trials that may improve the management of this cancer.

## Figures and Tables

**Figure 1 diagnostics-13-01949-f001:**
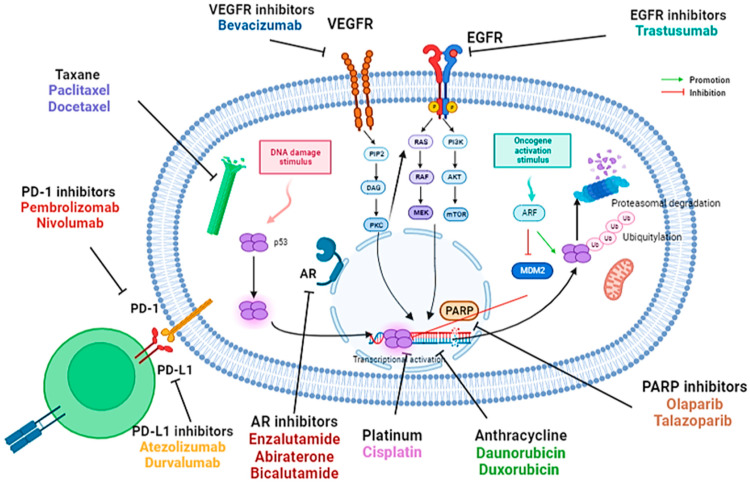
Different therapeutic targets in tumor cells in current clinical treatments in triple-negative breast cancer.

**Table 1 diagnostics-13-01949-t001:** Dedicated treatment options according to TNBC subtype [22].

Subtype	Characteristics	Treatment Options
Luminal AR (LAR),	AR expression	AR-targeted therapy
Immunomodulatory (IM)	Active immune response	PD-1/PD-L1 inhibitors
Basal-like 1 (BL1)	BRCA mutation	PARP inhibitors
Basal-like 2 (BL2)	Expression of myoepithelial markers such as epidermal growth factor receptor (EGFR) or cytokeratin 5/6 (CK5/6)	mTOR inhibitors
Mesenchymal/mesenchymal stem-like (M/MSL)	Epithelial–mesenchymal transition represented by E-cadherin negativity	PI3K inhibitors

**Table 2 diagnostics-13-01949-t002:** Ectopic expression of some microRNAs in TNBC.

miRNA	Expression Change	Targets	Functions	References
miR-9	Upregulated	CHN1	Inhibition of epithelial-to-mesenchymal transition (EMT). High expression associated with worse disease-free survival.	[139]
miR-10b	Upregulated	HOXD10	Induction of cell migration and invasion. High expression associated with lymph node metastases.	[140,141]
miR-21	Upregulated	PDCD4, PTEN, HIF1α, TIMP3, TM1	Enhancement of cell proliferation and invasion. High expression associated with poor prognosis.	[142,143]
miR-29	Upregulated	TTP	Activation of metastatsis. Expression inversely correlated with mammosphere stemness potential. No significant prognostic association.	[144]
miR-93	Upregulated	Unclear	Promotion of cell proliferation, invasion, and metastasis. Involvement in the process of EMT in endometrial carcinoma and breast cancer cells.	[145,146]
miR-145	Downregulated	MMP11, Rab27a	Inhibition of cell invasion through post-transcriptional regulation of gene targets.	[147]
miR-182	Upregulated	*PFN1*, FOXF2	Promotion of cell proliferation and invasion.	[148,149]
miR-199a-5p	Downregulated	CDH1, ZEB1, TWIST	Inhibition of EMT, cell migration, invasion, and tumor growth.	[150]
miR-200 family	Downregulated	EPHA2, ZEB1/2, SOX2, CD133, TWIST, CDH1, XIAP	Inhibition of migration and invasion. Promotion of differentiation in an undifferentiated mammary epithelial cell line. Strongly associated with increased chemoresistance.	[132,135,151]
miR-206	Downregulated	CORO1C, TM4SF1	Repression of tumor proliferation, migration, and invasion.	[152,153]
miR-211-5p	Downregulated	SETBP1	Suppression of tumor cell proliferation, invasion, migration, and metastasis.	[154]
miR-221	Upregulated	E-cadherin	Promotion of breast cancer progression.	[129,155]
miR-361-5p	Downregulated	RQCD1	Suppression of migration and invasion.	[122]
miR-603	Downregulated	eEF2K	Inhibition of cell proliferation, invasion, and tumorigenesis.	[156]
miR-4306	Downregulated	SIX1 Cdc42 VEGFA	Inhibition of cell proliferation, migration, and invasion.	[157]

**Table 3 diagnostics-13-01949-t003:** Summary of TNBC biomarkers.

Biomarkers	Gene	Localization	Alteration Type	Drug	Biomarker Effect	References
BRCA1/2	*BRCA1*	17q21.31	Sporadic mutations Epigenetic inactivation	PARP inhibitors	Therapeutic	[57,58]
	*BRCA2*	13q13.1				[57]
EGFR	*EGFR*	7p11.2	Overexpression	Tyrosine kinase inhibitors (EGFR-TKIs)	Therapeutic	[23,24]
FGFR	*FGFR1*, *FGFR2*, *FGFR3*, *FGFR4*	8p11.23	Amplification	FGFR inhibitors	Therapeutic/ preventive therapy	[29,30]
VEGFR	*VEGFR1/* *VEGFR2/* *VEGFR3*	6p21.3	Overexpression	Anti-VEGF antibody	Therapeutic	[24,38,39,42]
P53	*TP53*	17p13.1	Substitution	-	Prognostic	[45,46,50]
BCL2	*BCL2*	18q21.33	Anti-apoptotic	Anthracycline	Prognostic	[56,58]
Topoisomerase	*TOP1*	20q12	Amplification	Topo II inhibitors	Therapeutic	[67]
	*TOP2A*	17q21.2				[68]
PTEN	*PTEN*	10q23.31	Frameshift mutations	-	Prognostic	[69,74]
PD-1/PD-L1	*PDCD1*	2q37.3	-	PD-L1 inhibitors	Prognostic	[78]
	*CD274*	9p24.1				[79]
Notch pathway	*NOTCH1*	9q34.3	Gain of function (10%) In-frame deletion (13%)	-	Prognostic	[94]
	*JAG1*	20p12.2	-			[98]

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
