# Peer review of "New Biomarkers and Treatment Advances in Triple-Negative Breast Cancer"

_diagnostics, 2023, doi:10.3390/diagnostics13111949_

Round 1
Reviewer 1 Report
Review article includes very extensive research on all the types of biomarkers used for the diagnosis of TNBCs.
Sentences are very long and complex making the language hard to understand.
I will suggest authors to re-frame the sentences grammatically and make it easy to understand.
Otherwise paper has good amount of information and interesting to the reader.
Author Response
I am pleased to resubmit, on behalf of the co-authors, the revised version of the paper “diagnostics-2168288” entitled “New biomarkers and treatment advances in triple negative breast cancer.”, as well as the comments’ responses. We are thankful to the Reviewers and Editor-in-chief for the constructive comments that helped to considerably improve the quality of the manuscript.
We have carefully revised the present paper in the light of your suggestions.
Reviewer 2 Report
Congratulation for the Authors!
This is an excellent review of molecular pathology of triple negative breast cancer, that is the hopeless type of all. Systematic overview of targetable molecular changes, can sign druggable subtypes. This paper is a kind of guideline for breast specialist. Hope that more and more sub-subtype of TNBC would be treated as good results as HR+ ones. For the future I advice that radiomics should keep in parallel with these molecular genetic subtypes.
Author Response
I am pleased to resubmit, on behalf of the co-authors, the revised version of the paper “diagnostics-2168288” entitled “New biomarkers and treatment advances in triple negative breast cancer.”, as well as the comments’ responses. We are thankful to the Reviewers and Editor-in-chief for the constructive comments that helped to considerably improve the quality of the manuscript.
Reviewer 3 Report
Reviewer Comments
The review article, “New biomarkers and treatment advances in triple negative breast cancer” by Hejjioui et al., reports up-to-date knowledge of biomarkers in TNBC in conjunction with therapeutic approaches. Overall, it is well organized and provide knowledge for recent advances in TNBC biology and therapy. Before publication, followings are recommended.
Major comments
1. A table with summary of biomarkers will be very helpful.
2. Overall, sentences are too long to read. English editing to improve readability is needed. For example,
1) In the introduction, the first sentence is too long to read.
2) 7th paragraph of p. 3.
And there are many typos.
3. In 7th paragraph of p.2, the recent advances of targeted therapies, including PARP inhibitors, should be described in brief. Please see references such as PMID: 33801977 & PMID: 32869671.
4. In 2nd paragraph of p. 3, also brief description of recent targeted therapies in TNBC should be added.
5. In 6th paragraph of p. 3, it is not true that “with no targeted drug yet approved for TNBC”.
Minor comments
1. In 8th paragraph of p. 2, there is no reference.
2. In 2nd paragraph of p. 3, there is no reference.
3. In the title of Table 1
1) The first word should be started with capital letter.
2) The colon should be removed,
4. There is no reference to the first paragraph of p. 4.
5. The abbreviation should be referred once. For example, it is not necessary to use “Epidermal growth factor receptor (EGFR)” in the 2nd paragraph of p. 4, since author indicated the meaning of EGFR in the 2nd paragraph of p. 3. Please check the use of abbreviation in the entire manuscript.
6. The name of human protein should be expressed in upper case. Please see https://academic.oup.com/molehr/pages/Gene_And_Protein_Nomenclature. For example, Src-STAT3 in the 2nd paragraph of p. 4 should be changed to SRC-STAT3. Please check this through the manuscript.
7. For targeting EGFR, please see a recent paper, PMID: 34207383.
8. For targeting mtp53, there is an example: PMID: 23846322.
9. For epigenetic modifications, such as DNA methylation, there is an example: PMID: 29137356.
<The End>
Author Response
I am pleased to resubmit, on behalf of the co-authors, the revised version of the paper “diagnostics-2168288” entitled “New biomarkers and treatment advances in triple negative breast cancer.”, as well as the comments’ responses. We are thankful to the Reviewers and Editor-in-chief for the constructive comments that helped to considerably improve the quality of the manuscript.
We have carefully revised the present paper in the light of your suggestions and replied to each of your concerns below.
1- A table with summary of biomarkers has been added.
2- According to several articles, therapies targeting mTP53 and DNA methylation are still at the in vitro stage, therefore not approved by the FDA. We have tried in our article to focus on approved drugs.
All suggested revisions have been made.
Round 2
Reviewer 3 Report
Reviewer Comments
Authors performed revision of the manuscript accordingly. However, there are still some errors.
Major comments
1. Use exact definition of TNBC throughout the manuscript: immunohistochemical no expression of ER and PR, and no amplification of HER2. Oversimplified definition may lead misunderstanding.
2. Thorough English editing is required especially in the newly added sentences.
3. As mentioned earlier, throughout the manuscript, protein names should be expressed with upper cases. For example, use RAS and RAF instead of Ras and Raf, respectively.
4. The description of VEGF genes is quite confusing. Are VEGFA-to-E proteins encoded by a VEGF gene located on the 6p21.3? Does VEGF 165 mean VEGFA165?
5. Please add references in the Table 3.
Minor comments
1. Please exactly refer the chromosomal location of the EGFR gene.
<The End>
Author Response
Dear Editor,
Thank you for giving us the opportunity to submit a revised draft of our manuscript entitled New biomarkers and treatment advances in triple negative breast cancer to MDPI diagnostics journal. We appreciate the time and effort that you and the reviewer have dedicated to providing your valuable feedback on our paper. We are very grateful to the reviewer for their insightful comments and their high quality and constructive reviews of our manuscript. It is our belief that the manuscript is substantially improved after making the suggested edit.
In this revised version, first, we have kept all the corrections made in the manuscript by the reviewer, which we found very interesting. In addition, we did our best to address all comment raised by the reviewer. We have highlighted the changes within the manuscript (Green color).
Here is a point-by-point response to the reviewer comments and concerns.
- Comment 1: Used exact definition of TNBC throughout the manuscript.
Response: Thank you for pointing this out. We agree with this comment. Therefore, modified the definition of TNBC throughout the manuscript.
- Comment 2: English editing.
Response: the authors have done their best to improve language of the manuscript especially in the newly added sentences (spelling and grammatical errors have been corrected).
- Comment 3: Protein names should be expressed with upper cases.
Response: we have made the requested changes throughout the manuscript (the proteins in upper case and genes in italic).
- Comment 4: The description of VEGF
Response: we have added a paragraph with more details.
- Comment 5: Add references in the Table 3.
Response: we have added the references in table 3.
Comment 6: Refer the chromosomal location on the EGFR gene.
Response: we have added the exact chromosomal location of the EGFR gene (7p11. 2).
Thank you very much again for giving us the chance to revise this paper. If you have any questions, please let us know.
Sincerely,
EL HEJJIOUI BRAHIM
Round 3
Reviewer 3 Report
Authors appropriately addressed all the questions except for VEGF. Please add further description on VEGF genes. See PMID: 15693956.
Author Response
In this revised version, first, we have kept all the corrections made in the manuscript by the reviewer, which we found very interesting. In addition, we did our best to address all comment raised by the reviewer. We have highlighted the changes within the manuscript (Green color).
- Comment: The description of VEGF
Response: We have detailed more the VEGF gene part with more references.
Thank you very much again for giving us the chance to revise this paper. If you have any questions, please let us know.
